



# Technical note: Can Visual Gauges Trained on Biased Contact-based Gauge Data Accurately Estimate River Stage?

Ze Wang[1], Heng Lyu[1,2], Guangtao Fu[3], Chi Zhang[1]

[1]School of Hydraulic Engineering, Dalian University of Technology, Dalian, Liaoning, China

[2]NingBo Institute of Dalian University of Technology, Ningbo, Zhejiang, China

[3]Faculty of Environment, Science and Economy, University of Exeter, Exeter, UK

*Correspondence to:* Heng Lyu ([lyuheng@dlut.edu.cn](mailto:lyuheng@dlut.edu.cn))

**Abstract.** Water stage variations significantly influence biochemical and hydrological processes within river networks. River camera, with its ease of deployment and low cost, has emerged as a promising tool for water stage estimation, enabling efficient water stage interpretation from images via deep learning (DL). However, a critical challenge is the requirement of accurate water stage data for DL training, which often have biases caused by sedimentations, floating debris or water flow impacts associated with contact-based gauge observations. Previous studies have overlooked the influence of gauge data errors in real-world applications. This study introduces an imaging-based water stage estimation framework that addresses hidden errors in gauge station measurements for training DL models. The framework adopts a multi-task learning paradigm, using erroneous gauge stage data as labels and incorporating water pixel ratios automatically extracted from images to constrain model estimation ranking. Based on training loss, a thresholding method then filters error-free data to retrain an unbiased model. This framework is tested on images and bubble-gauge stage data from the Minturn River, Greenland, spanning 2019 to 2021. The results obtained show the framework successfully identified a gauge offset event on July 29, 2021, and mitigated an average water stage observation error of approximately 0.6 meters thereafter. Moreover, the trained DL model revealed water stage fluctuations under low-flow conditions that gauge observation could not reflect. This study implies that integrating contact and non-contact observations is a robust approach for river stage measurement.



# 1. Introduction

Rivers are conduits between terrestrial and aquatic environments, mediating hydrological processes and biochemical transports (Mosley, 2015; Whitworth et al., 2012). Observing river flow dynamics can enhance comprehension of river evolution mechanisms, aiding in the development of adaptive water management plans and early warning systems for extreme hydrological events (Kreibich et al., 2022; Lane, 2017; Trenberth et al., 2014). As a typical indicator of river flow dynamics, water stage data are essential for unraveling complex exchange fluxes among different hydrological components (Etter et al., 2018; Van Wesemael et al., 2019; Yamazaki et al., 2012). Therefore, water stage has become one of the primarily observed hydrological variables for rivers.

River water stage observations can be categorized into contact-based and non-contact-based methods, each associated with distinct sources of observational error. Traditional river gauges, as a representative of contact-based methods, commonly use float, pressure, or vibrating wire sensing as a foundational component for water stage measurement (Kinzli et al., 2016; Loizou & Koutroulis, 2016; Majdalani et al., 2019). Constrained by the contact-mode, these gauges are prone to interference from sedimentation, floating debris, and drift under flow impacts, compromising the accuracy and consistency of observation (Di Baldassarre & Montanari, 2009). Non-contact observation methods, exemplified by remote sensing via satellites such as Envisat and Jason-2, can supplement water stage data for in-situ river gauges (Frappart et al., 2006; Papa et al., 2012). Due to orbital constraints, atmospheric interference, and limited sensor revisit frequencies, satellite and airborne optical techniques are restricted by relatively long revisit intervals and susceptibility to cloud cover. The resulting water stage data typically exhibit temporal resolutions of 5–16 days and estimation errors exceeding 0.5 meters (Grimaldi et al., 2016; Yan et al., 2015).

River cameras, as an arising form of non-contact, near-range remote sensing techniques, can mitigate the impact of river flow and objects on measurement (Dolcetti et al., 2022). By providing real-time visualization of river conditions, river cameras can yield high temporal resolution water stage data, supplementing traditional gauge observations (Spasiano et al., 2023). The effectiveness of river camera observations depends on the accuracy of water stage extraction from the images. A series of studies installed white poles, colored bars, or QR codes as fiducial markers within the camera's view, then adopted computer vision techniques to track pixel value changes of these markers, allowing the estimation of water stages (Herzog et al., 2022; Noto et al., 2022; Tauro et al., 2022). These methods share a common reliance on contact-based markers placed within river channels, with cameras acting as around-the-clock observers for continuous water stage monitoring.



Another commonly used imaging-based method involves segmenting water pixels within each image using traditional computer vision methods or deep learning (DL) models (Akiyama et al., 2020; Erfani et al., 2022; Lopez-Fuentes et al., 2017). Segmented water masks can be transformed using photogrammetric techniques and then overlaid onto the topography

of river channels to derive scalar water stage values (Sermet & Demir, 2023). The accuracy of water stage data obtained through this method depends on both waterbody segmentation and terrain data quality. Factors such as suspended solids, water transparency, and surface reflection can affect the accuracy of river topographical measurements derived from LiDAR scanning. These errors propagate through the workflow and ultimately impact the overlay analysis of water masks and terrain data for river stage estimation (Awadallah et al., 2023; Hilldale & Raff, 2008).

Using water stage data observed by gauges as labels to train a DL model provides a novel approach for directly mapping images to water stages, enabling accurate water stage estimations from new images (Vandaele et al., 2023; Vanden Boomen et al., 2021). This approach avoids the need for deploying additional reference objects and collecting fine-scale terrain data. Previous studies have already validated the inherent fitting capabilities of DL-enabled image regression models for imaging-based water stage estimation across various rivers (Gupta et al., 2022). A commonly used convolutional neural network

architecture, ResNet18, was trained on gauge station data from six small streams in the United States, achieving a mean absolute error as low as 0.1 m. However, in practical scenarios, the gauge water stage data used for training may already contain errors stemming from water flow disturbances. As DL models trained under noisy labels are likely to learn biased mapping relationships (Northcutt et al., 2021; Yi et al., 2022), estimated water stages can be inaccurate, rendering them unable to produce robust water stage measurement based on river camera images. Notably, this issue has received limited

attention in prior research, and addressing gauge data bias remains a critical knowledge gap in achieving robust imaging-based observations.

To mitigate the impact of inherent errors in historical gauge data, this study proposes a novel framework that integrates a multi-task learning paradigm with a thresholding method to train a DL regression model, enabling accurate imaging-based water stage estimation. Concurrently establishing a mapping relationship between images and observed gauge data during

model training, we introduce an additional ranking loss function for constraining model parameter optimization. The ranking loss guides the DL model to ensure that the ordering of water stage estimations aligns with the sequence of water body pixel ratio estimated by the water segmentation models. A thresholding method is subsequently developed to adaptively detect moments of historical bias based on the training loss value for each image, allowing for the automatic identification of error-free historical gauge data to retrain a robust DL model.






## 2. Methodology

### 2.1 Overview of the framework

ShuffleNet, a classic lightweight convolutional neural network (X. Zhang et al., 2018), was selected as the backbone model for imaging-based water stage estimation. Through the use of depth-wise separable convolutions and channel shuffling,

ShuffleNet achieves high computational and memory efficiency (Ma et al., 2018). To apply the ShuffleNet model to provide accurate water stage estimations despite existing errors in historical gauge data, we developed a multi-phase framework incorporating multi-task learning and a thresholding method (Figure 1).

In the first phase, multi-task learning was implemented by training the ShuffleNet model with both gauge stage sequences and imaging-based water pixel ratio sequences as labels. The former served as the direct target for model fitting, though it

contains potential errors, while the latter was used to further constrain the fitting process by enforcing consistency in the output values' ordering relationship with SOFI (Static Observer Flooding Index, SOFI). Based on water masks automatically extracted from images using a water segmentation model (Z. Wang et al., 2024), SOFI was calculated as a ratio of water pixel to the total pixel of the image. By using dual labels, the impact of biases in the gauge data can be mitigated.

The Second phase is designed to detect gauge bias. A thresholding method was applied to the sequence of differences

between model predictions and gauge stage to identify potential gauge bias occurrences, segmenting the sequence into two subsequences, representing data before and after the gauge bias occurrence. A statistical test was then conducted to determine whether the former significantly differed from the latter; if not, this indicated no significant error in the gauge stage sequences. Otherwise, a significant difference confirmed the presence of gauge bias, allowing the erroneous gauge stage data to be discarded, thus isolating only reliable data for further use. Finally, the error-free gauge stage subsequence,

along with the complete SOFI sequence, was used as labels to retrain ShuffleNet. Through the use of error-free data, a DL regression model capable of providing accurate water stage estimations thus can be yielded. Specific details of each component in the framework will be elaborated in subsequent sections.




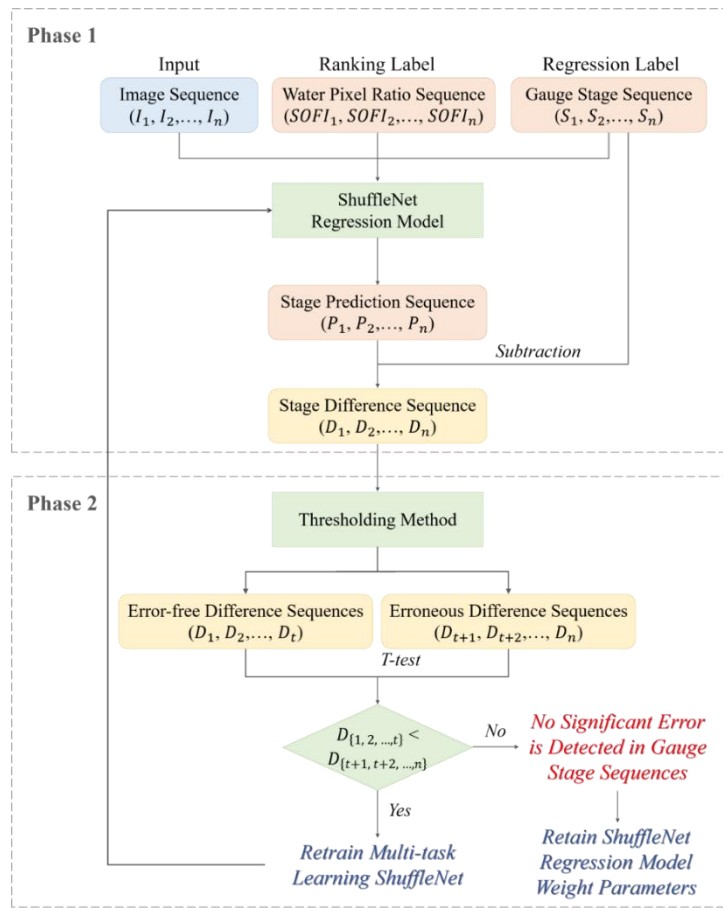

**Figure 1. The workflow of the framework designed to train a DL-enabled image regression model capable of providing accurate**
**water stage estimation under biased gauge data conditions.**

## 2.2 Multi-task learning paradigm

Multi-task learning involves solving multiple learning tasks simultaneously (Y. Zhang & Yang, 2021). By capturing the
common pattern hidden in the training signal of related tasks, MTL has the potential to extract more information from the
training data and enhance the model performance (Ruder, 2017). As shown in Figure 2 (a), this study formalized the model
training into a combination of regression and ranking tasks. Initially, water stage predictions were compared with gauge
stage for supervising the regression task. Meanwhile, margin loss was computed by pairing all images in the mini-batch and
testing whether their rankings adhered to the ranking of SOFI values (Figure 2 (b)).





The ShuffleNet model jointly learned the regression task and the ranking task, by defining the total loss function as:


$$L_{total} = L_{reg} + \lambda L_{rank} \tag{1}$$

$L_{reg}$ and $L_{rank}$ are the regression and ranking loss, respectively, and $\lambda$ is the weighting parameter to balance the contributions of the two terms. For $L_{reg}$, we used the Mean Squared Error (MSE) function:

$$L_{reg} = (v - v_{gt})^2 \tag{2}$$

where $v$ represents the model prediction and $v_{gt}$ is the ground truth value of the river water stage. The ranking loss, $L_{rank}$, is

computed as:

$$L_{rank} = \max\left(0, -v_{sofi}^{rank}(v_1 - v_2)\right) \tag{3}$$

where $v_1$ and $v_2$ represent the model prediction for the two images in a pair, and $v_{sofi}^{rank}$ represents the ground truth ranking for the pair referring to the SOFI, where "+1" means the SOFI value for *image 1* is higher, and "-1" means the SOFI value for *image 2* is higher.

The ranking loss serves as a regularization term, preventing overfitting of the regression objective with the aid of SOFI derived from river camera images, especially when inherent errors exist in gauge stage observations due to gauge bias. The weight λ balances the regression and ranking tasks, ensuring it is sufficiently large for regularization but not excessively high to compromise water stage estimation accuracy. We showed the influence of varying λ in Section 3.2.

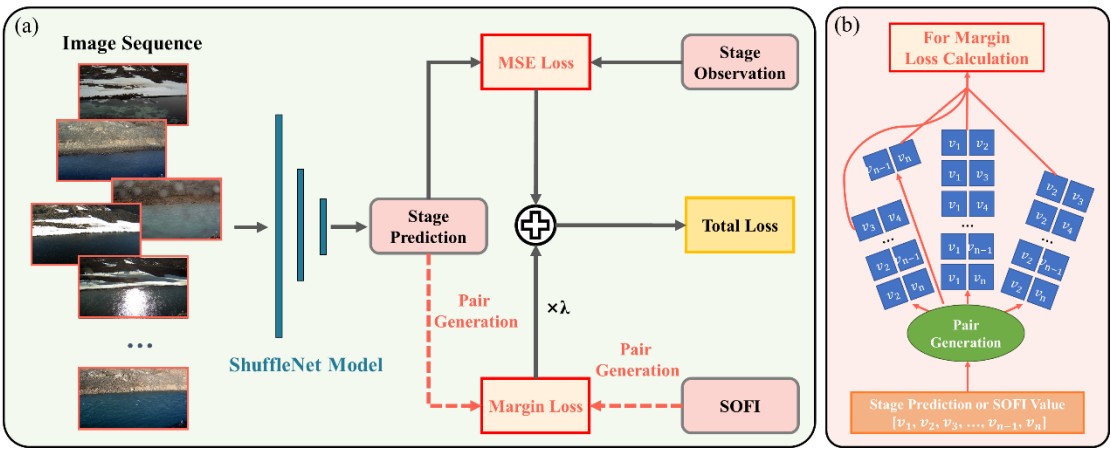

**Figure 2. Multi-task learning paradigm. (a) The illustration of the training framework that combines both regression and ranking tasks; (b) the workflow of margin loss calculation for supervising the ranking task.**



## 2.3 Thresholding method

After iterative training under the multi-task learning paradigm, the stage difference sequence can be calculated by subtracting the gauge stage from the model prediction. Subsequently, a thresholding method was used to detect the occurrence time of the most significant gauge bias within the inspected period, based on which erroneous stage data were removed.

Initially, the Jenkins Natural Break method (Jenks, 1967) was used to divide the stage difference sequence into two groups: values exceeding the binary threshold were classified as significant errors (*True*), likely attributable to gauge offset, while those below the threshold were identified as insignificant errors (*False*) due to random variations. While significant errors do not exclusively occur following gauge bias caused by flowing debris or flow impact—since a minority of images may exhibit significant errors due to poor visual quality—the majority of cases still display a temporal clustering of significant and non-significant errors. Accordingly, errors occurring before and after the gauge bias were assumed as *False* and *True*, respectively. We then aimed to optimize the estimated gauge bias occurrence time to maximize alignment between the time-sequenced labels obtained from the thresholding method and the assumed labels. Specifically, the objective is to achieve the highest F1 score between the two label sequences.

F1 score is a commonly-used metric for evaluating the classification performance. It considers both the precision and recall of the model to compute a single score. Precision ($P$), Recall ($R$), and F1 score ($F$) are calculated as follows:

$$P = \frac{TP}{TP+FP} \tag{4}$$

$$R = \frac{TP}{TP+FN} \tag{5}$$

$$F = \frac{2 \times P \times R}{P+R} \tag{6}$$

Where TP is the true positive rate representing the proportion of samples that are actually positive and judged as positive, while TN is the true negative rate representing the proportion of samples that are actually positive but are judged as negative. Similarly, FP and FN denote the false positive rate and false negative rate, respectively. Here, the positive and negative samples were time points labeled as *True* and *False*.

To confirm that the identified significant errors are systematic errors caused by physical contact, a T-test was applied. Only if a statistically significant comparative relationship was observed between the difference subsequences before and after the





gauge bias occurrence would the post-bias gauge stage sequence be discarded, retaining only the pre-bias subsequence and the complete SOFI sequence to retrain ShuffleNet under the multi-task learning paradigm for accurate water stage estimation.

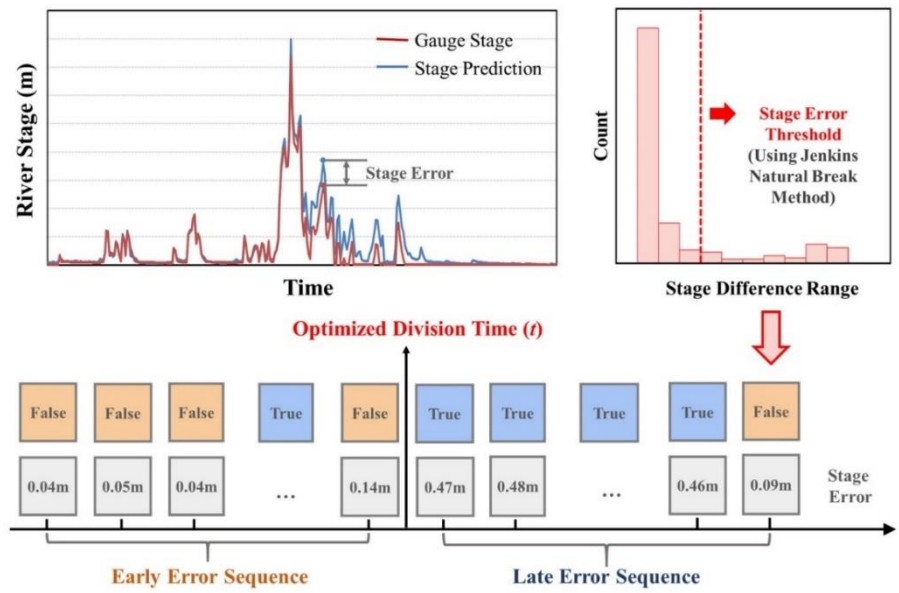

**Figure 3. The illustration of the thresholding method for inferring the optimized division time.**

## 3. Study area and experimental setup

### 3.1 Study area

The Minturn River was selected as the study area to evaluate the effectiveness of the proposed framework due to its concurrent collection of river stage gauge data and time-lapse camera images on the main stem of the river. As shown in Figure 5 (a), the Minturn River is a major proglacial river that drains a ~2,800 km² supraglacial catchment, originating from a large grounded lobe of the Greenland Ice Sheet in Inglefield Land, Northwest Greenland (Li et al., 2022; Yang et al., 2019). The gauging site was positioned ~15 km downstream of the ice edge on the Minturn River, enabling the collection of runoffs from the largest river within the proglacial zone.

As shown in Figure 4 (b), the instrument monitoring network was established by Goldstein et al (2023) and consists of an in-situ bubble-gauge and two time-lapse camera systems. The camera positioned on the west bank was used in this study,





capturing images under adequate ambient light conditions. Its oblique field-of-view is directed across the river to monitor the right (east) bank of the river channel. A total of 288, 377, and 317 images were obtained between July 12 and December 17 2019, between June 21 and December 31 2020, and between January 1 and September 17 2021, respectively, and their

corresponding river stage gauge data were also obtained.

The bubble-gauge experienced significant errors after July 29, 2021, when the protective conduit was struck, resulting in the shearing-off of the exit orifice, likely caused by a large upstream boulder impacting the conduit. Such systematic biases in the bubble-gauge data make it an ideal candidate for testing the effectiveness of our framework. Additionally, at the gauge site, the Compact CF Bubbler of the bubble-gauge becomes exposed at low flow, while observable flow remains in the river

channel. As the compact bubbler can only measure water stage when submerged, it measures atmospheric pressure instead of water stage when the river level falls below it.

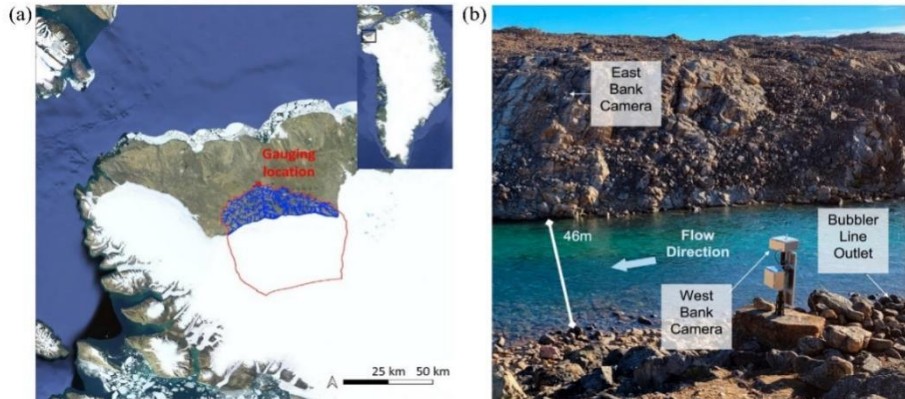

**Figure 4. (a) Study area and gauging location. The red lines delineate the boundary of the Minturn River watershed, while the blue**
**lines represent the tributaries of the Minturn River. (b) On-site photo of the Minturn River gauging station, including bubble-**
**gauge and time-lapse cameras. Source: Goldstein et al (2023).**

**3.2 Experimental setup**

Three datasets with varying data volumes, excluding or including systematic bias, were constructed: the 2019 data, the data
from 2019 to 2021, and the data from 2019 to 2021. The framework was applied separately to all three datasets to evaluate its performance under varying data conditions. For each dataset, the original gauge stage data were used as labels to train the



model, while the corrected gauge stage data, obtained through the overlay analysis of water masks and terrain data for the corresponding period, served as a benchmark to evaluate the framework's effectiveness in detecting gauge bias and assess ShuffleNet model's estimation performance.

When training the ShuffleNet model under the multi-task learning paradigm, the number of training iterations was set to 20, the learning rate was set to 0.001, and the mini-batch size was set to 8. Additionally, to investigate the impact of the parameter λ used to balance the regression and ranking tasks on model training, nine scenarios were set (λ=0, 1, 5, 10, 15, 20, 25, 30, 35), and the iterative changes in MSE loss were visualized for each scenario to determine the optimal λ value. As illustrated in Figure 5, an increase in λ was associated with a corresponding rise in the initial MSE loss value, and the loss

did not consistently converge to zero throughout the iterative process. This suggests that multi-task learning effectively mitigates overfitting to the inherently erroneous gauge stage data. Beyond a λ value of 10, further increases in λ did not result in significant changes in the MSE loss variation. In addition, continuously increasing λ may cause the model to prioritize ranking performance over accurate regression, thereby limiting its ability to constrain water stage estimates within reasonable numerical ranges. Consequently, a λ value of 10 was selected for subsequent analysis.

Furthermore, the Spearman correlation coefficient (de Winter et al., 2016), were applied to preliminarily evaluate the consistency of the SOFI sequence and gauge stage sequence prior to model training. In addition, Root Mean Square Error (RMSE) and Mean Absolute Error (MAE) were adopted to quantify the gaps between different water stage sequences.

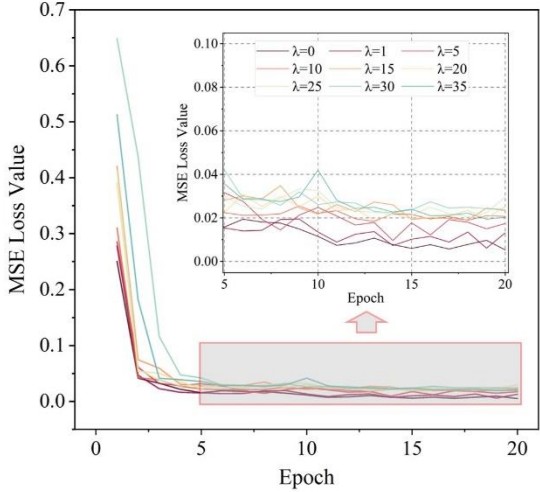

**Figure 5. MSE loss value curve with different λ values**




## 4. Results

### 4.1 Comparative relationship between target variables in multi-task learning

Accurate water segmentation and reliable SOFI sequence forms the foundation upon which the incorporation of ranking task can mitigate the impact of erroneous gauge stage on model training. Based on the water masks automatically extracted by the
water segmentation model, SOFI for each image was calculated and compared with the corresponding gauge stage.

As shown in Figure 6 (a), SOFI calculated from images collected in 2019 and 2020 highly correlated to the gauge stage, with Spearman rank correlation coefficients of 0.99 and 0.95, respectively. Due to systematic errors from gauge bias, the correlation between SOFI and gauge stage was lower in 2021, with a Spearman rank correlation coefficient of 0.87.

The SOFI exhibited consistency with the gauge stage, yet the two displayed a non-linear relationship (Figure 6 (b)).
Although inferring the gauge bias occurrence time based on direct comparison of these variables is challenging, the reliability of SOFI calculated through the method used in this study, along with its alignment with gauge stage trends, qualifies it to serve as an effective label for supervising the ranking task in the multi-task learning paradigm.





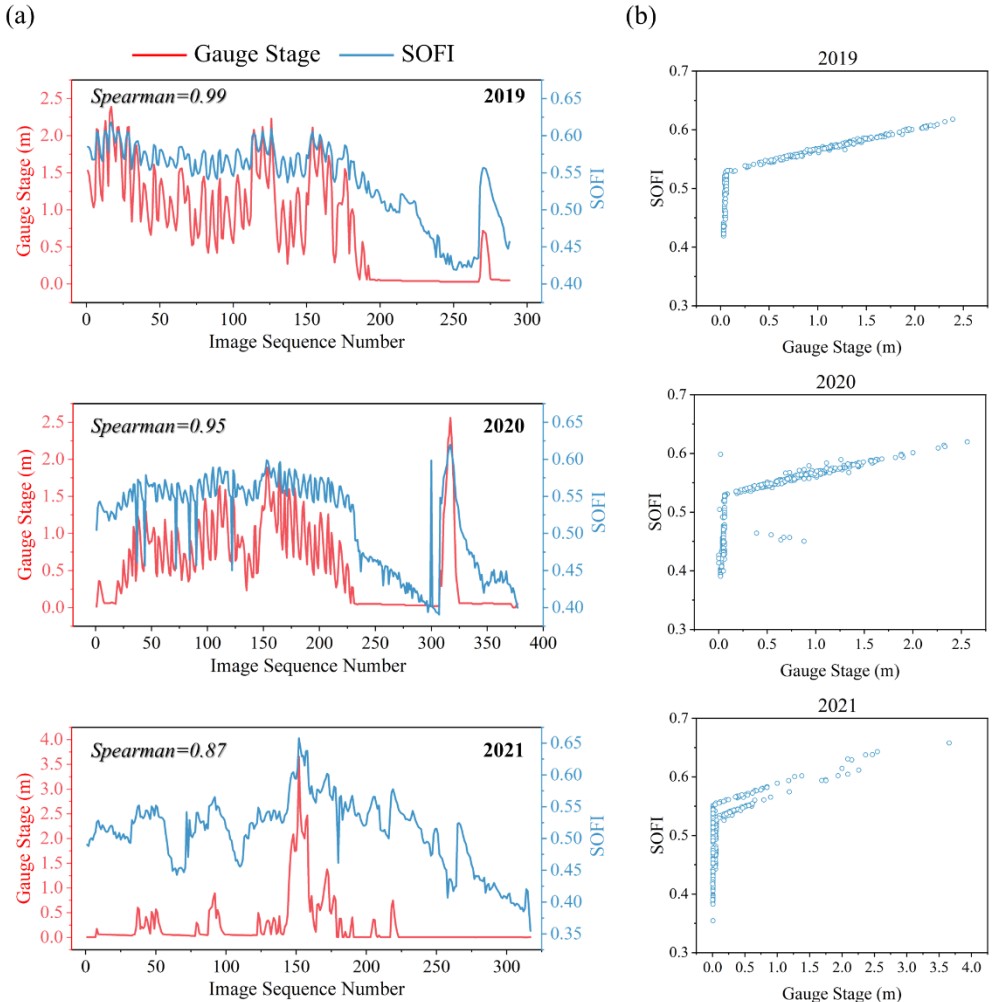

**Figure 6. The comparative relationship between gauge stage and SOFI in years 2019 – 2021. (a) Historical data of gauge stage and**

**SOFI in different years and their (b) scatter plots.**

## 4.2 Detection of erroneous gauge stage

After iterative training under the multi-task learning paradigm, the differences between the stage estimated by the ShuffleNet model from the image and the gauge stage were computed. Based on these differences, the thresholding method was applied

to infer the time when the gauge bias occurred.



For the complete dataset covering 2019 to 2021, the Jenks Natural Breaks method established a threshold of 0.21 m, classifying all absolute stage errors as either *True* or *False*. Among these, 924 time points were categorized as *False*, while 58 time points were categorized as *True* (Figure 7). Following this, the optimal segmentation time point was determined by maximizing the F1 score, leading to the identification of the gauge bias occurrence time as 0:00 a.m. on July 30th, 2021,

consistent with the actual event. A T-test was then used to verify the gauge bias occurrence determined by the thresholding method. A p-value less than 0.01 confirmed the presence of gauge bias.

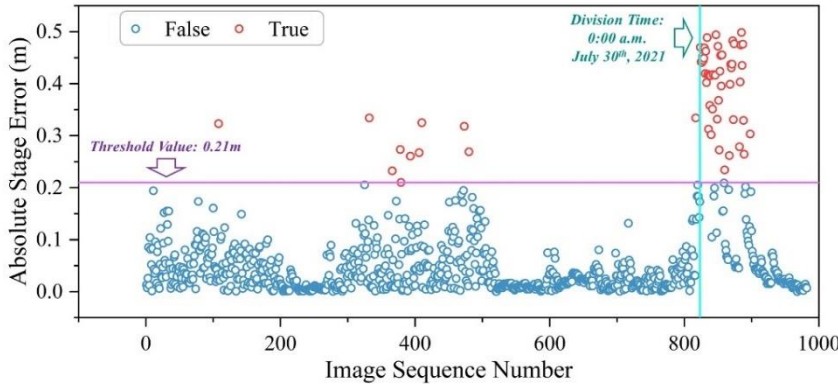

**Figure 7. The temporal sequence of absolute stage error and their corresponding labels.**

Unlike the 2019–2021 dataset, the other two datasets—the 2019 data and the 2019–2020 data—did not contain errors induced by significant gauge bias and were used for comparison to demonstrate the robustness of the framework. The same process was applied to both datasets as with the complete dataset, estimating the threshold value and conducting a T-test to assess differences in absolute stage error before and after the inferred gauge bias occurrence. As shown in Figure 8, in the absence of inherent significant errors, the estimated threshold values were relatively small (2019: 0.06 m; 2019–2020: 0.05

m), and no significant difference in errors was observed before and after, indicating that no gauge bias existed, consistent with the actual scenario. This validates the robustness of the framework.





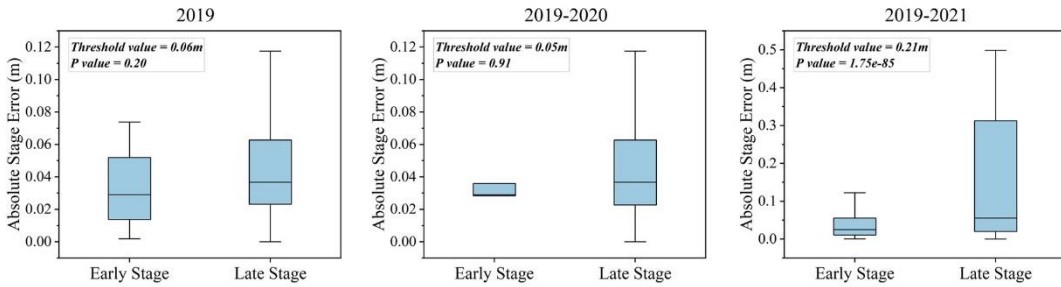

**Figure 8. Absolute stage errors of both early and late stages for three datasets.**

## 4.3 Retraining effectiveness

After detecting the gauge bias occurrence, we used the error-free gauge stage data along with the complete SOFI sequence from 2019 to 2021 to retrain the ShuffleNet model under the multi-task learning paradigm, aiming to establish a robust mapping relationship between the image data and the unbiased river water stage.

As illustrated in Figure 9 (a), after model retraining, ShuffleNet predicted river water stage consistent with the gauge stage for images captured before the occurrence of gauge bias, with RMSE and MAE below 0.05m. For the period after gauge bias occurrence, significant corrections were made to the gauge stage, resulting in RMSE and MAE of 0.56m and 0.57m, respectively, which are consistent with the gauge bias magnitude determined by previous studies based on cameras and Lidar scanners (Goldstein et al., 2023).

In comparison, results from single-task learning or multi-task learning without re-training led to complete overfitting or remained partially affected by erroneous labels, failing to fully mitigate the impact of errors in gauge stage data. This highlights the necessity of model re-training (Table 1). Furthermore, as shown in Figure 9 (b), the model could further extrapolate water stage values into negative ranges relative to the original coordinate system, and effectively depict water stages below the bubble-gauge installation point, thereby supplementing observations of low-flow conditions.




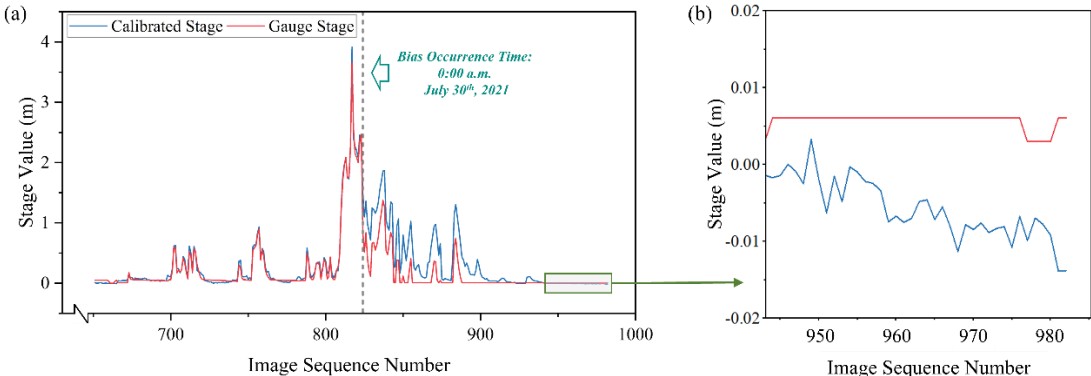

**Figure 9. Effectiveness of estimations under low flow conditions. (a) The temporal sequence of the calibrated stage after model retraining; (b) example illustration of stage variations for low flow conditions.**

**Table 1.** Comparative performance of training strategies on river water stage estimation

| | RMSE (m) | | MAE (m) | |
|---|---|---|---|---|
| Training Strategy | Phase without Bias (Ideal Value: 0) | Phase with Bias (Ideal Value: 0.5) | Phase without Bias (Ideal Value: 0) | Phase with Bias (Ideal Value: 0.5) |
| Single-task Learning | 0.03 | 0.04 | 0.02 | 0.03 |
| Multi-task Learning (pre-training) | 0.05 | 0.10 | 0.04 | 0.08 |
| Multi-task Learning (post-training) | 0.05 | 0.57 | 0.04 | 0.56 |


## 5. Discussion

In this study, the SOFI sequence obtained from river camera and the water stage data collected by bubble-gauge were concurrently used to train a DL regression model. The original single regression task of mapping images to water stages was extended to integrate the regression and ranking tasks into a multi-task learning paradigm. In the presence of systematic

errors in the gauge stage data, the ranking task supervised by SOFI was introduced as a regularization item (Tian & Zhang,

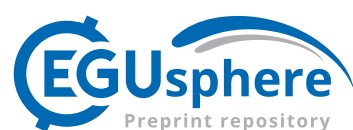

2022). With the assistance of relatively more robust non-contact data, model parameter optimization can be effectively guided, thus preventing overfitting to inherently biased data. From data perspective, the multi-task learning can be viewed as a form of data assimilation, whereby multiple data sources are integrated during training, facilitating mutual calibration. The approach of using multi-task learning for data assimilation has already been applied to monitoring other hydrological
variables such as flood depth and water quality (Chaudhary et al., 2020; Zhe et al., 2023). It can effectively address challenges including the absence of fine-grained data and data uncertainties.

The integration of multiple data sources through multi-task learning relies on the reliability of at least some of them or on the premise that the error types of each data source are distinct. For instance, in our study, the stability of the non-contact monitoring mode and the accuracy of the water segmentation algorithm render SOFI relatively robust. The reliability of
camera data underpins the final framework's ability to accurately infer gauge bias occurrences and rectify historical data. Moreover, it should be noted that that our framework only concentrates on addressing systematic errors stemming from gauge biases. Other sporadic errors caused by periodic hydraulic impacts can be effectively distinguished through the simple application of a T-test. These errors, characterized by random distribution patterns, can potentially be mitigated directly by leveraging the bias inductive capacity of deep learning (DL) (Baxter, 2000), obviating the need for specific processing.

Moreover, there are still some limitations that need to be addressed. Firstly, the study inadequately considered the possibility of multiple gauge biases. The bubble-gauge deployed on the Minturn River, used in our study, experienced only one significant bias during 2019-2021. To address the occurrence of multiple biases, future work can implement sliding time windows with shorter intervals to sequentially process the entire time series or adopt unsupervised clustering methods to automatically identify and separate clusters (Khan et al., 2014), thus enabling the automatic recognition of multiple gauge
biases. Meanwhile, the framework will also be tested on more diverse datasets with different types and levels of error to validate its performance. Furthermore, when river cameras are deployed in environments prone to displacement by strong winds or wildlife, applying image quality control and calibration techniques such as image registration (Zitova & Flusser, 2003) may be needed.



## 6. Conclusion

This study has presented that DL regression models can yield reliable water stage estimations from river camera images, even when training data contain errors stemming from floating debris or hydraulic impact spikes. A novel framework integrating multi-task learning and a thresholding method was developed, and was able to effectively identify gauge stage errors and mitigate their impact on model training, establishing a robust mapping from images to water stage.

The framework was applied to bubble-gauge data and river camera images collected in the Minturn River, Greenland, from 2019 to 2021. It accurately detected the occurrence of gauge bias on July 29, 2021, and corrected an average water stage observation error of approximately 0.6 m thereafter. Additionally, the trained DL model based on the framework can further reveal water stage variations under low-flow conditions below the bubble-gauge installation point. Furthermore, the framework was validated to focus specifically on systematic errors, rather than over-identifying and processing minor random errors, demonstrating adaptability to varying data conditions.

Overall, this study presents a novel approach to integrating biased gauge data with river camera images for robust, imaging-based river stage measurement. Future work should address multiple occurrences of gauge bias, potential displacements of river cameras, and extend the framework's application to a wider range of river conditions for comprehensive evaluation.

### Acknowledgement

We acknowledge the financial supports from China Ministry of Science and Technology (2024YFC3213000), Open Research Fund Program of State Key Laboratory of Hydraulics and Mountain River Engineering (SKHL2415), National Natural Science Foundation of China (U2240204), and Natural Science Foundation of China (52309005).

In addition, we would like to share the study's interesting intriguing origin. This study began by chance. Dr. Wang Ze tried to train an image-to-water-level deep-learning model using publicly available data from Greenland. A subset of the data stubbornly resisted fitting, which puzzled him. After digging into the associated literature, he discovered that those records were indeed erroneous. This realization raised a broader question: since many hydrologic observations contain errors, must they be excluded from AI training? Pursuing this idea, the present study was born.





**Data availability statement**

The river image and gauge datasets used in the paper can be accessed through DOI: 10.5281/zenodo.7951337.

**Disclosure statement**

The authors declare that they have no conflict of interest.



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
