# Peer review of "Technical note: Can Visual Gauges Trained on Biased Contact-based Gauge Data Accurately Estimate River Stage?"

_EGUsphere, 2025_

## Referee Comment (RC2)

**Line-by-line comments**

Wang et al. (2025) - Technical Note: Can Visual Gauges Trained on Biased Contact-based Gauge Data Accurately Estimate River Stage?

**Major issues:**

L. 152: If I am not totally mistaken, it's quite the opposite. True negative is the proportion of samples that are correctly judged as negative! The authors need to make sure this was only confused in the text and not also in their analysis.

L. 173: Why is there a months-long pause in 2020? I think the authors should address this.

L. 249 (Fig.8): what is "early stage" and "late stage"? This is not defined anywhere and does not seem obvious. I do not understand where the late stage in dataset 2019 and 2019-2020 come from if there was no gauge failure.

L. 259: in my opinion one cannot rely on such an extrapolation below gauge zero without any gauging data supporting this. This might work well in one location and completely fail in another one, solely based on river bed geometry. Just because the model provides values below zero does not mean one can trust them.

**Minor issues:**

L. 2: Title: I think the title could be chosen a bit more carefully to accurately reflect the content of this work: only 1 visual gauge is used and it's a case study with only one event. I propose something like: "Estimating river stage with a visual gauge trained on biased gauge data - a case study"

L. 50: "contact-based" does not seem like the right adjective for markers. They aren't contact-based sensors even though they are technically in contact with the water. I'd suggest simply going with ".. reliance on markers placed within river..."

L. 88 ff: adding the word "sequence" after each data type seems unnecessary and confuses the reader (also applies to Fig.1).

L. 94: Since the term "gauge bias" is a core idea of this publication, it should be defined more precisely. If understood correctly the authors use this term to address singular events that significantly offset the stage measurements.

L. 108: the authors should add the abbreviation MTL in brackets right after mentioning Multi-task learning for the first time.

L. 213: I am missing a more detailed description of how the water segmentation was performed. This seems to be a core part of this work.

L. 220 ff: I suggest rephrasing this sentence and splitting it into several sentences. Shorter and more understandable would increase the readability significantly.

L. 234: To many people it is highly unclear whether 0:00 am means midnight or noon. Instead I'd suggest using 24h time format or at least noon/midnight.

L. 249 (Fig.8): the intuitive interpretation of the three graphs suggests that all three datasets are similar. Only when looking at the y-scale of the third plot one may notice that they are of different scale. The authors should consider using the same scale for all three plots.

**Grammar, trivialities:**

L. 11: I suggest using the plural "river cameras" here, as we are talking about a general type of instrumentation rather than one specific camera.

L. 286: "that that"

L. 302: here the authors could consider using "data" as a singular here to improve the readability of the sentence: "... even when training data contains errors..."

---

## Author Comment (AC1)

**Dear Reviewer 1:**

We appreciate your careful reading and critical reflections on our manuscript. We value the concerns raised, but respectfully hold a different perspective on several key points. Below, we address each comment in detail, presenting our reasoning and the corresponding revisions to be made to the manuscript.

**Comment 1:**

The most critical concern lies in the overall suitability and motivation for an approach that learns the absolute stage directly from images. The reliance on on-site gauge data for training at every new location significantly limits its utility, particularly for ungauged catchments, which are the primary target for innovative remote sensing techniques. As gauged catchments already possess well-established, high-accuracy stage measurement methods, the practical added value of this camera-based approach for these sites is questionable. Also, there are already studies discussing the potential and limits of directly learning the stage from images, which are not mentioned in this study (e.g. Vanden Boomen et al., 2021). Furthermore, the risk is high that the approach is highly sensitive to any movements (internal or external geometry) of the camera setup. Such movements would likely necessitate a complete re-learning of the model, which is a significant practical limitation and is not adequately addressed in the current work. Finally, the authors' premise that obtaining accurate stage data is a critical challenge for all DL-based camera gauges is debatable. For approaches relying on photogrammetry, the stage data serves only as a reference, not as the primary input for the AI-model, thereby mitigating this "critical challenge." A stronger, more refined motivation for this specific DL-only approach is needed.

**Response to Comment 1**

We thank the reviewer for this insightful comment regarding our technical route. In response, we would like to offer complementary perspectives on the intrinsic value of image-based river monitoring.

First, we acknowledge that our approach currently requires physical gauge data at a new site for initial model training. However, the data dependency is limited to the early stage of deployment. For a specific site, during the initial co-existence period of cameras and gauges at the early phase, the latter can serve as reference data to help AI models learn and stabilize. Once the physical gauges reach the end of their lifespan, a well-trained camera-AI system can replace their function at a low cost, enabling continuous observation. With the continuous increase in the number of river camera and physical gauge observation sites, as well as the accumulation of image-water level pairs, DL models that directly establish image-to-water-stage mappings without relying on additional camera parameters or environmental information such as terrain provide a foundation for developing one-fit-all water stage estimation models. In contrast to other methods such as photogrammetric approaches, these models are

independent of site-specific auxiliary data, thereby demonstrating strong generalizability and the potential to be directly extended to ungauged rivers without reliance on physical gauges or in-situ topographic surveys.

Second, we believe that our method possesses a certain degree of robustness to variations in the visual observation environment. As the reviewer pointed out, factors such as camera displacement and geometric variation are important sources of potential uncertainty that can affect the robustness of image-based water level estimation. Taking photogrammetric estimation as an example, this method requires the extraction of water masks and their subsequent overlay with topographic data. When the camera viewpoint changes but the pre-established projection coordinate system is still used, substantial biases in the derived water level can occur. In contrast, the DL framework we adopted, which directly maps images to water stages, can inherently mitigate such effects and maintain robustness. DL models can automatically extract high-level features that capture relative spatial relationships between different objects, such as water bodies and riverbanks, rather than depending on the absolute position of any single target. Consequently, minor camera displacements exert limited influence on the results, while larger shifts can be effectively corrected using modern camera systems equipped with automated calibration and pan-tilt-zoom (PTZ) mechanisms capable of dynamically compensating for geometric variations in real time.

Overall, we consider our chosen technical route to be feasible in terms of generalizability and robustness. Within this framework, the quality of water-stage values obtained from physical gauges—used as training labels for the AI model—is critical. However, influenced by contact-based measurement errors, these labels often contain either *random but minor* deviations or *systematic and substantial* biases. Random noise caused by turbulence or backflow can typically be attenuated during deep-learning parameter optimization. In contrast, systematic errors—such as those arising from sediment accumulation or rapid riverbed changes—necessitate targeted correction strategies. Accordingly, our study's core, innovative contribution is the introduction of a multi-task learning framework that uses the water-pixel proportion from images as an auxiliary label to mitigate systematic errors from physical gauges.

In the revised manuscript, we will refine and integrate these perspectives into the *Introduction* and *Discussion* sections to better articulate the rationale and significance of the proposed deep-learning-enabled, image-based water level estimation approach. Regarding the reviewer's comment on the citation of previous studies, we agree that a more in-depth discussion of their limitations would be beneficial. Therefore, we will incorporate and critically discuss these studies in the revised version to more clearly highlight both the potential and the limitations of prior research in this area. We also recognize that the previous phrasing, "obtaining accurate stage data is a critical challenge for all DL-based camera gauges", was inaccurate. This challenge pertains specifically to the technical route used in our study, rather than to all DL-based

camera gauge approaches in general. The statement will be revised in the manuscript to reflect this clarification.

**Comment 2:**

The paper utilizes pixel information from segmented images to provide relative stage information but lacks sufficient discussion on the segmentation process itself. This is a significant omission, especially since several established studies (e.g., Eltner et al., 2021; Zamboni et al., 2025, Moghimi et al., 2024) already perform this kind of water segmentation for stage measurement, and the potential for segmentation errors and their influence on the multi-task learning is not discussed at all. Furthermore, the review fails to include relevant, state-of-the-art photogrammetric approaches that use water segmentation (e.g., Blanch et al., 2025). Given that the study site appears highly suitable for these methods, a direct comparison and justification for choosing the DL-only approach is necessary. Also, the achieved accuracy, appearing to be in the decimeter (dm) range, is not competitive with the centimeter (cm) accuracy demonstrated by other camera gauge studies, particularly those using robust photogrammetric methods (e.g., Eltner et al., 2021, Erfani et al., 2023, Blanch et al., 2025). Therefore, also the title of the manuscript is misleading because I think, the achieved accuracies cannot be described accurate. Finally, the approach involves combining two loss functions, which necessitates the fine-tuning of the lambda value. This introduces a hyperparameter that must be manually tuned, complicating the model's reliability and generality.

**Response to Comment 2:**

First, we understand the reviewer's focus on segmentation-based approaches. However, it is important to clarify that the image segmentation module is not the innovation of this study. In our previous work (Wang et al., 2025), we have already proposed and validated an advanced water-body segmentation algorithm, which demonstrated state-of-the-art performance on representative datasets.

The algorithm integrates a domain-specific DL-based water segmentation model with a foundation model, Segment Anything Model (SAM). Compared with the DL-enabled semantic segmentation models used by Erfani et al. (2023) and Blanch et al. (2025), our method exhibits stronger generalizability and lower dependence on local data. In contrast to the SAM-based segmentation approaches proposed by Moghimi et al. (2024) and Zamboni et al. (2025), our method further incorporates an additional DL module as an automated prompter, which enhances usability and facilitates automated deployment. Without any local fine-tuning, the proposed segmentation approach achieved an IoU exceeding 0.9 across four river camera stations in Tewkesbury, UK, demonstrating sufficient performance for generating auxiliary labels within the multi-task learning process of this study.

Overall, the principal contribution of the present work lies in the design of a multi-task learning framework that integrates relative water-level features derived from the segmentation task with the direct water-level regression task. This joint formulation effectively mitigates the impact of biased contact-based gauge data during model training, thereby improving the robustness of water level estimation. A detailed discussion of the segmentation algorithm itself would divert attention from this core methodological innovation — the proposed multi-task learning structure. Therefore, the detailed description of the segmentation method will not appear in the *Introduction*. Instead, following the reviewer's suggestion, we will add appropriate citations and a concise description of the segmentation algorithm in the *Methods* section to justify the use of our internally developed segmentation approach. Moreover, we do not deny the value of photometric approaches that combine water segmentation with topographic projection. In fact, in this study, the ground-truth water stage data were obtained using such a photogrammetric method, as the physical gauge measurements were affected by contact-based biases.

In addition, we would like to clarify a misunderstanding regarding the reported accuracy. The accuracy of our results is expressed in centimeters. This unit convention follows the same precision definition used in previous studies conducted at the same site, ensuring comparability and methodological consistency. Nevertheless, we concur with the reviewer that the use of "accurate" in the title may convey a subjective impression. The title will be revised to more clearly highlight the methodological focus and problem addressed, with the effectiveness of the approach objectively demonstrated through the presented results.

Regarding the hyperparameter that balances the two tasks, we have conducted dedicated experiments and ablation analyses to examine their effects. The results and parameter recommendations are reported in the manuscript and can serve as a practical reference for future applications (Line 195 - Line 204):

"When training the ShuffleNet model under the multi-task learning paradigm, the number of training iterations was set to 20, the learning rate was set to 0.001, and the mini-batch size was set to 8. Additionally, to investigate the impact of the parameter  $\lambda$  used to balance the regression and ranking tasks on model training, nine scenarios were set ( $\lambda$  =0, 1, 5, 10, 15, 20, 25, 30, 35), and the iterative changes in MSE loss were visualized for each scenario to determine the optimal  $\lambda$  value. As illustrated in Figure 5, an increase in  $\lambda$  was associated with a corresponding rise in the initial MSE loss value, and the loss 200 did not consistently converge to zero throughout the iterative process. This suggests that multi-task learning effectively mitigates overfitting to the inherently erroneous gauge stage data. Beyond a  $\lambda$  value of 10, further increases in  $\lambda$  did not result in significant changes in the MSE loss variation. In addition, continuously increasing  $\lambda$  may cause the model to prioritize ranking performance over accurate regression, thereby limiting its ability to constrain water stage estimates within reasonable numerical ranges. Consequently, a  $\lambda$  value of 10 was selected for subsequent analysis."

**Comment 3:**

The suggested automatic detection of gauge errors appears effective only for very strong and obvious errors. It is unclear why an established statistical approach would not be equally or more effective for this task. The authors apply an automatic post-processing/filtering step to refine the training data, assuming the error resides in the stage data and not the camera imagery. This assumption needs stronger justification. The lack of provided code is a serious concern, particularly for a technical note. This does not comply with the FAIR principles, which are essential for research reproducibility.

**Response to Comment 3:**

The contact-based biases in physical gauges can generally be divided into random and systematic components. DL models inherently possess a strong capability to smooth out stochastic noise during the optimization process. In our study, an automatic detection module was specifically developed to identify and mitigate significant systematic errors.

Regarding the assumption issue, the strength of the multi-task learning framework lies in its ability to balance and integrate errors across different tasks, thereby enhancing the learning performance of each task simultaneously. It is not necessary to impose a prior assumption about the superiority of one data source over another, and indeed, we have not tried to make such assumption in this study. Furthermore, minor geometric or imagery-related shifts caused by slight camera movement or displacement do not adversely affect the learning performance of DL models, as we have explained in our response to comment 1.

With regard to traditional statistical approaches, these methods typically rely on causal relationships, in which additional meteorological variables — such as precipitation and temperature — are incorporated to correct or explain variations in stage data. In the absence of such external drivers, however, it becomes challenging to distinguish genuine hydrological dynamics from observation errors. In contrast, our approach is grounded in correlative relationships derived directly from camera imagery, providing a visually supported validation pathway that operates independently of meteorological inputs. While future extensions may integrate these meteorological factors, the current framework is designed to address challenges that traditional statistical methods cannot effectively resolve without them. Therefore, the two approaches are conceptually distinct and not directly comparable. The *Discussion* section of the revised manuscript will further elaborate on the theoretical distinctions between causal and correlative modeling paradigms

Finally, we appreciate the reviewer's valuable suggestion regarding code accessibility. We will make the full source code publicly available and include the corresponding

repository link in the revised manuscript to facilitate reproducibility and future research use.

**Reference**

Blanch, X., Grundmann, J., Hedel, R., & Eltner, A. (2025). AI Image-based method for a robust automatic real-time water level monitoring: A long-term application case. https://doi.org/10.5194/egusphere-2025-724

Erfani, S. M. H., C. Smith, Z. Wu, E. A. Shamsabadi, F. Khatami, A. R. J. Downey, J. Imran, and E. Goharian. 2023. "Eye of Horus: A Vision-Based Framework for Real-Time Water Level Measurement." Hydrology and Earth System Sciences 27 (22): 4135–4149. https://doi.org/10.5194/ hess-27-4135-2023.

Moghimi, A., M. Welzel, T. Celik, and T. Schlurmann. 2024. "A ComparativePerformance Analysis of Popular Deep Learning Models and Segment Anything Model (SAM) for River Water Segmentation in Close-Range Remote Sensing Imagery." Institute of Electrical and Electronics Engineers Access 12:52067–52085. https://doi.org/10.1109/ACCESS.2024.3385425

Wang, Ze, Heng Lyu, Yanqing Guo, Shun'an Zhou, and Chi Zhang. 2025. How to Use General AI for Task-Specific Applications: A Case Study of Monitoring Water Level Trends with River Cameras. Environmental Modelling & Software 192:106550. https://doi.org/10.1016/j.envsoft.2025.106550.

Zamboni, P. A. P., Blanch, X., Marcato Junior, J., Gonçalves, W. N., & Eltner, A. (2025). Do we need to label large datasets for river water segmentation? Benchmark and stage estimation with minimum to non-labeled image time series. International Journal of Remote Sensing, 46(7), 2719–2747. https://doi.org/10.1080/01431161.2025.2457131

---

## Author Comment (AC2)

Dear Daniel,

We sincerely appreciate your thoughtful comments and your recognition of the scientific value of our work. Meanwhile, we fully understand the concerns you raised regarding the methodology, and we would like to take this opportunity to provide further clarification to alleviate these concerns. The relevant parts will be reinforced and refined in the revised manuscript.

Comment 1:

Even though the work is well motivated, the method fails to convince, that it can be generalised. It relies on gauging data for training, which rules out ungauged streams.

Response to Comment 1:

As you rightly pointed out, our approach does require paired images and gauging data for initial training. However, we believe this method has greater potential for generalization in the long term compared with other imaging-based water stage observation techniques.

Existing approaches for interpreting water stage information from river camera images generally fall into two categories. The first category uses water segmentation algorithms to extract water masks, which are then overlaid on high-resolution surveyed terrain to retrieve the water stage. This method requires in-situ terrain surveying with LiDAR each time a new river-camera station is deployed. The second category, which is adopted in our study, directly learns the mapping relationship between images and water stage using deep learning regression models. As more river camera stations are deployed, the deep learning model can increasingly learn to extract higher-order features by jointly considering background environmental cues, enabling direct water stage estimation with no reliance on terrain measurements.

Therefore, our approach can potentially support independent and real-time deployment of river cameras, making it more generalised in the long term, and offering a pathway toward application in ungauged streams when pretrained on sufficiently diverse image-stage pairs.

Comment 2:

Furthermore the authors base this methodology on a case study using just a single camera at one location and including only one (very pronounced) gauge failure event. No claim can be made whether this method holds up under different conditions.

Response to Comment 2:

We fully understand your concern regarding the limited number of test sites, however, we would like to emphasize that the proposed method has been systematically evaluated at two complementary levels.

First, under conditions where no systematic gauge bias is present, a robust method should not mistakenly interpret small random measurement noise as a systematic error, nor should it degrade otherwise reliable stage observations. To evaluate this aspect, we constructed three temporal subsets (2019, 2019-2020, and 2019-2021), representing scenarios with and without pronounced gauge failures. The consistent performance across these datasets (as shown in Fig. 8) demonstrates that our method does not over-detect random fluctuations and exhibits a certain degree of generality under normal conditions.

Second, when a true systematic bias does occur, the method should be able to accurately localize its onset in time and apply an effective overall correction to the affected period. This capability is demonstrated in the presented pronounced gauge failure case in our study. We acknowledge that, within the currently available dataset, only one such well-documented systematic bias event is available. Therefore, while this case provides clear proof-of-concept evidence, further validation across multiple independent failure cases will be addressed in the future work.

Taken together, the current experiments provide objective evidence for both the general robustness of the method under normal conditions and its effectiveness in correcting true systematic biases under documented failure scenarios, while also clearly indicating directions for further strengthening its applicability.

Comment 3:

Finally, the overall accuracy is not able to compete with the current state-of-the-art sensors.

Response to Comment 3:

We agree that a rigorous assessment of accuracy against state-of-the-art sensing technologies is essential. In our study, the water stage estimates obtained by the proposed method were quantitatively compared with those derived from a conventional computer-vision-based workflow based on waterline extraction and surveyed terrain overlay. The results show a clear improvement relative to the original gauging records affected by systematic bias, achieving a mean absolute error (MAE) of 0.04 m after correction.

This level of accuracy is comparable to, and in some cases better than, previous state-of-the-art image-based river stage estimation studies, such as Vanden Boomen et al. (2021, MAE ≈ 0.07 m) and Eltner et al. (2021, MAE ≈ 0.05 m):

*Eltner, A., Bressan, P. O., Akiyama, T., Gonçalves, W. N., & Marcato Junior, J. (2021). Using deep learning for automatic water stage measurements. Water Resources Research, 57(3), e2020WR027608. https://doi.org/10.1029/2020WR027608*

*Vandaele, R., Dance, S. L., & Ojha, V. (2021). Deep learning for automated river-level monitoring through river-camera images: An approach based on water segmentation and transfer learning. Hydrology and Earth System Sciences, 25(8), 4435–4453. https://doi.org/10.5194/hess-25-4435-2021*

In the revised manuscript, we will add extra quantitative comparison to further contextualize the achieved performance. We also sincerely welcome the reviewer to recommend any additional recent state-of-the-art references for further comparison, and we will be pleased to include them accordingly.

Comment 4:

In some parts the scientific quality is hard to judge, since the individual steps are only documented partially. The segmentation process used to obtain the water pixel percentage is not further described, despite it being one of the two input variables for training the model. Even though the process is adopted from a previous publication by some of the authors, it should be described in more detail, as it's a major potential source of errors. The same applies to the model training and retraining process. The authors report little on what exactly was done and to which effect, giving a bare minimum of information and providing no supplementary information.

Response to Comment 4:

Regarding the documentation of individual methodological steps, we are pleased to provide additional details. Among them, the water segmentation method used in this study is a technique we previously published. Initially, we were concerned that elaborating on this part might distract readers from the core innovation of the present work, the multi-task learning framework. However, we have noticed readers' interest in the segmentation method and acknowledge that segmentation exactly plays a critical role in the overall framework. Therefore, we will include more detailed descriptions and performance illustrations of the segmentation algorithm in the revised manuscript. Likewise, additional details on the training and retraining procedures will also be added.

Below are our detailed responses to your major issues, minor issues, and grammatical comments.

Major issues:

Major Issue 1:

L. 152: If I am not totally mistaken, it's quite the opposite. True negative is the proportion of samples that are correctly judged as negative! The authors need to make sure this was only confused in the text and not also in their analysis.

Response to Major Issue 1:

Thank you for pointing out this mistake. The definition of true negative in the manuscript was indeed incorrectly stated due to a writing error. In the revised manuscript, we will correct it to:

"*TN represents the number of samples that are actually negative and correctly predicted as negative.*"

We would also like to clarify that this error was limited to the textual description only. The quantitative analysis itself was not affected, because all evaluation metrics (Precision, Recall, and F1 score) were computed using the built-in functions from *scikit-learn* module in Python, which handle the definitions of TP, TN, FP, and FN internally and correctly. Therefore, the results remain valid.

Major Issue 2:

L. 173: Why is there a months-long pause in 2020? I think the authors should address this.

Response to Major Issue 2:

We appreciate your careful observation. The data used in our study were obtained directly from the open-source dataset provided in Goldstein et al. (2023). The authors do report a several-month gap in camera images during early 2020, but they do not provide an explicit explanation for the cause of this interruption. However, the data gap occurs well before the period in which the systematic gauge error took place. Therefore, the missing portion does not affect the identification or analysis of the systematic bias that represents the core focus of our study.

We will add a brief clarification in the revised manuscript to acknowledge this data gap and cite the original source.

*Goldstein, S. N., Ryan, J. C., How, P. R., Esenther, S. E., Pitcher, L. H., LeWinter, A. L., Overstreet, B. T., Kyzivat, E. D., Fayne, J. V., & Smith, L. C. (2023). Proglacial river stage derived from georectified time-lapse camera images, Inglefield 360 Land, Northwest Greenland. Frontiers in Earth Science, 11(June), 1–11. https://doi.org/10.3389/feart.2023.960363*

Major Issue 3:

L. 249 (Fig.8): what is "early stage" and "late stage"? This is not defined anywhere and does not seem obvious. I do not understand where the late stage in dataset 2019 and 2019-2020 come from if there was no gauge failure.

Response to Major Issue 3:

We apologize for the confusion caused by the terms "early stage" and "late stage." In our manuscript, these terms are intended to denote the time periods before and after the potential gauge-bias occurrence, respectively.

As described in Lines 155-158 of the manuscript, a potential split point is only considered a true gauge-bias event if the errors in the two subsequences exhibit a statistically significant difference based on a two-sample T-test. When such a significant difference exists, the post-bias subsequence is treated as biased and discarded; otherwise, the detected fluctuation is regarded as random noise rather than a systematic error:

"*To confirm that the identified significant errors are systematic errors caused by physical contact, a T-test was applied. Only if a statistically significant comparative relationship was observed between the difference subsequences before and after the gauge bias occurrence would the post-bias gauge stage sequence be discarded, retaining only the pre-bias subsequence and the complete SOFI sequence to retrain ShuffleNet under the multi-task learning paradigm for accurate water stage estimation.*" (Line 155-158)

Therefore, the datasets "2019" and "2019-2020" are also divided into "early" and "late" subsequences, but the split points in these datasets are not identified as genuine systematic bias events, because their before–after errors are not statistically distinguishable.

To avoid ambiguity, we will revise the caption of Fig. 8 to explicitly explain that the "early stage" and "late stage" refer to the periods before and after the candidate gauge-bias timing, regardless of whether the candidate point is ultimately validated as a true systematic error.

Major Issue 4:

L. 259: in my opinion one cannot rely on such an extrapolation below gauge zero without any gauging data supporting this. This might work well in one location and completely fail in another one, solely based on river bed geometry. Just because the model provides values below zero does not mean one can trust them.

Response to Major Issue 4:

We agree that extrapolating water-stage estimates below the gauge-zero reference without any supporting gauging data involves inherent uncertainty. As you correctly pointed out, riverbed geometry can influence the validity of such extrapolations.

However, we would like to clarify that the negative stage values generated by the model are not intended to be interpreted as physically reliable absolute measurements that can be directly applied in practice. Instead, they are used solely to demonstrate the potential of the vision-based approach to capture water stage variations below the camera installation reference level within the image-based frame of reference. Whether these sub-zero estimates are quantitatively consistent with the true physical water stage remains an open question and requires dedicated validation using independent observations in future studies.

Overall, we appreciate your rigor and acknowledge that the physical reliability of such extrapolations requires further validation against independent reference measurements. In the revised manuscript, we will explicitly clarify this issue in the Discussion section to avoid any potential misunderstanding.

Minor issues:

Minor Issue 1:

L. 2: Title: I think the title could be chosen a bit more carefully to accurately reflect the content of this work: only 1 visual gauge is used and it's a case study with only one event. I propose something like: "Estimating river stage with a visual gauge trained on biased gauge data - a case study"

Response to Minor Issue 1:

We agree that the title should more accurately reflect the scope of our study, We appreciate your proposed title and will revise the manuscript title accordingly, or adjust it in a similar manner to better highlight the scope of this work.

Minor Issue 2:

L. 50: "contact-based" does not seem like the right adjective for markers. They aren't contact-based sensors even though they are technically in contact with the water. I'd suggest simply going with ".. reliance on markers placed within river..."

Response to Minor Issue 2:

We agree that "contact-based" is not appropriate here and will revise it to "reliance on markers placed within the river."

Minor Issue 3:

L. 88 ff: adding the word "sequence" after each data type seems unnecessary and confuses the reader (also applies to Fig.1).

Response to Minor Issue 3:

We intentionally use the term "sequence" to highlight that the analysis is based on time-series error data. We acknowledge the concern, but we prefer to retain the term for conceptual clarity.

Minor Issue 4:

L. 94: Since the term "gauge bias" is a core idea of this publication, it should be defined more precisely. If understood correctly the authors use this term to address singular events that significantly offset the stage measurements.

Response to Minor Issue 4:

We agree that a clearer definition is needed. In our study, "gauge bias" refers specifically to singular systematic events that cause a significant and sustained offset in stage measurements. We will clarify this definition in the revised manuscript.

Minor Issue 5:

L. 108: the authors should add the abbreviation MTL in brackets right after mentioning Multi-task learning for the first time.

Response to Minor Issue 5:

Thanks. We will add the abbreviation "MTL" after its first occurrence.

Minor Issue 6:

L. 213: I am missing a more detailed description of how the water segmentation was performed. This seems to be a core part of this work.

Response to Minor Issue 6:

As noted earlier, we will add additional details on the water segmentation method in the revised manuscript.

Minor Issue 7:

L. 220 ff: I suggest rephrasing this sentence and splitting it into several sentences. Shorter and more understandable would increase the readability significantly.

Response to Minor Issue 7:

We agree with this suggestion and will rephrase and split the sentence to improve clarity and readability in the revised manuscript.

Minor Issue 8:

L. 234: To many people it is highly unclear whether 0:00 am means midnight or noon. Instead I'd suggest using 24h time format or at least noon/midnight.

Response to Minor Issue 8:

We agree and will revise the time expression to an unambiguous 24-hour format.

Minor Issue 9:

L. 249 (Fig.8): the intuitive interpretation of the three graphs suggests that all three datasets are similar. Only when looking at the y-scale of the third plot one may notice that they are of different scale. The authors should consider using the same scale for all three plots.

Response to Minor Issue 9:

We understand the concern about visual comparability. However, applying the same y-axis scale to all three plots would substantially compress the variations in Fig. 8a and Fig. 8b, making it difficult to visually discern the differences between their early and late subsequences. For this reason, we prefer to retain separate y-axis scales.

Grammar, trivialities:

L. 11: I suggest using the plural "river cameras" here, as we are talking about a general type of instrumentation rather than one specific camera.

L. 286: "that that"

L. 302: here the authors could consider using "data" as a singular here to improve the readability of the sentence: "... even when training data contains errors..."

Response to "Grammar, trivialities" issues:

We will thoroughly check and correct these grammatical issues in the revised manuscript